# Application of the Advanced Surface Modification Process to the ASME Code Case for Sections III and XI of Nuclear Power Plants

**Sungwoo Cho [1],\*, Hyun-Uk Hong [2], Nicholas Mohr [3], Marc Albert [3], John Broussard [4], Auezhan Amanov [5] and Young-Sik Pyun [5],\***

1   Doosan Heavy Industries and Construction, Changwon 51711, Korea
2   Changwon National University, Changwon 51140, Korea; huhong@changwon.ac.kr
3   EPRI, Charlotte, NC 28262, USA; nmohr@epri.com (N.M.); MAlbert@epri.com (M.A.)
4   Dominion Engineering, Inc., Reston, VA 20191, USA; jbroussard@domeng.com
5   Department of Mechanical Engineering, Sun Moon University, Asan 31460, Korea; amanovtribo@gmail.com
\*   Correspondence: sungwoo1.cho@doosan.co.kr (S.C.); pyoun@sunmoon.ac.kr (Y.-S.P.)

**Abstract:** The advanced surface modification process is known as a promising solution to improve the performance of machine components and systems, especially for vehicles, nuclear power plants, biomedical device, etc. There have been several successful applications of water jet peening and underwater laser peening to nuclear components in Japan since 2001 which resulted in inspection and repair cost savings. The prerequisite condition for the application of the advanced surface modification process to nuclear power plants is the approval of the American Society of Mechanical Engineers (ASME) Code Case, so performance criteria and requirements (PCRs) in the ASME Code Case for repair and maintenance of nuclear power components are explained. A challenging project to apply advanced surface modification processes, such as ultrasonic nanocrystal surface modification and air laser peening to new nuclear power plants and new canisters, was created with the goal to develop a technical basis and the PCRs for ASME Section III (New Manufacturing). The results of this work will be an ASME Section III Code Case which is currently in progress. An initial draft of the new Code Case with the intermediate results of this work is introduced. Four kinds of advanced surface modification processes are explained and compared briefly.

**Keywords:** advanced surface modification process; water jet peening (WJP); under water laser peening (ULP); ultrasonic nanocrystal surface modification (UNSM); air laser peening (ALP); primary water stress corrosion cracking (PWSCC); dry canister; chloride-induced stress corrosion cracking (CISCC); ASME Code Case

## 1. Introduction

There have been successful applications of water jet peening (WJP) and under water laser peening (ULP) to the maintenance of nuclear components in Japan since 2001 as listed in Table 1 [1]. The prerequisite condition for the application of the advanced surface modification process to nuclear power plants in the USA, Korea, etc., is the approval of an ASME Code Case. So, two ASME Code Cases have been developed since November of 2011. "Alternate Examination Requirements and Acceptance Standards for Class 1 pressurized water reactor (PWR) Piping and Vessel Nozzle Butt Welds Fabricated with UNS N06082 or UNS W86182 Weld Filler Material with or without Application of Listed Mitigation Activities" was approved on 7 May 2014 and updated as Code Case N-770-4 in ASME Section XI, Division 1 [2]. "Alternative Examination Requirements for PWR Reactor Vessel Upper Heads with Nozzles Having Pressure-Retaining Partial-Penetration Welds" was approved on 7 October

2015 and updated as Code Case N-729-5 in ASME Section XI, Division 1 [3]. The application of WJP and ULP, which is called a surface stress improvement process or advanced surface modification process, is to reduce the likelihood of primary water stress corrosion cracking (PWSCC). Hence, the expected benefits are to increase safety, production, and service life. The USA Nuclear Regulatory Commission has approved inspection relief for the facilities which have applied the surface stress improvement process according to ASME Code Case N-729-5 or ASME Code Case N-770-4 and MRP-335-3A (Topical Report for PWSCC Mitigation by Surface Stress Improvement) [4]. The application of surface stress improvement on susceptible components is expected to have inspection and repair cost savings.

**Table 1.** Application of the advanced surface modification process in Japan.

| Process | Application Area | Power Plants (Cumulative) | Since |
|---------|------------------|---------------------------|-------|
| WJP | Bottom Mounted Nozzle; Inner Diameter, Outer Diameter, J–Groove, Reactor Vessel Inlet Outlet Nozzles | PWR 59 plants | 2001 |
| | Core Shroud, Guide Tubes, In-Core Monitor Housing, Control Rod Drive Housing, Jet Pumps, Sub-Tubes, etc. | BWR 17 plants | |
| ULP | BMN; ID, OD, J–Groove, RV Inlet Outlet Nozzles, Deluge nozzles | PWR 2 plants | 2004 |
| | Core Shroud, Guide Tubes, ICM Housing, CRD Housing, Jet Pumps, Sub-Tubes, etc. | BWR 8 plants | |

Another challenging project was started to apply advanced surface modification processes, such as ultrasonic nanocrystal surface modification (UNSM) and air laser peening (ALP), to ASME Code Case for Section III. The intent of this Code Case is to prolong the service life of new nuclear power plants and new canisters for more than 80 years and 120 years, respectively. The intermediate results of this project were reported by two papers in the 2019 ASME Pressure Vessels and Piping Division Conference PVP-2019. The first one is "New Code Case Development for the Mitigation of primary water stress corrosion cracking (PWSCC) and Chloride Induced Stress Corrosion Cracking (CISCC) in ASME Section III Components by Advanced Surface Stress Improvement Technology" [5], and the second one is "A Development of the Technical Basis for the New Code Case 'Mitigation of PWSCC and CISCC in ASME Section III Components by the Advanced Surface Stress Improvement Technology" [6].

In the current paper, these four kinds of advanced surface modification processes are introduced and compared briefly. Their PCRs are to be applied for repair and maintenance and which were registered in ASME Code Case for Section XI. The draft of the PCRs with the development of their technical basis for new components in ASME Code Case for Section III and an initial draft of ASME Code Case are explained.

## 2. Brief Explanation of Advanced Surface Modification Processes

The difference between a traditional peening process and an advance surface modification process is defined in ASME Code Case N-4422 [7]. Traditional peening is a process (e.g., shot peening, pneumatic needle gun) that physically deforms the material by cold working to control distortion. A surface stress improvement process, which is also called an advanced surface modification process, is a process that reduces the residual tensile stresses on the surfaces of welds and base material caused by welding or cold-working processes. Both WJP and ULP were used for mitigation in ASME Section XI Code Case N-729-6 and N-729-5, while UNSM and ALP are being evaluated to prepare a technical basis for "New Code Case Development for the Mitigation of PWSCC and CISCC in ASME Section III Components". The concepts, mechanisms, and application examples of WJP, ULP, and ALP were introduced in Electric Power Research Institute (EPRI) Reports MRP-267-1 and MRP-335-3. The concepts of WJP, ULP, and ALP are explained in those documents, and more detail on concepts,

mechanisms, and application examples of UNSM is needed.  Finally, a comparison of four major processes is summarized in Table 2 [1,4,8,9].

**Table 2.** Comparison of the four advanced surface modification process.

| Contents | WJP | ULP | ALP | UNSM |
|---|---|---|---|---|
| Mechanism | Shockwave | | | Resonance/Continuous Contact |
| Source of Impact | Cavitation Bubble | Laser Ablation | | Solid Ball/Tip |
| Source of Energy | Water Jet | Pulse Laser | | Ultrasonic Vibration |
| Contact Pressure and Impulse | ~10 GPa Kinetic | ~12 GPa Kinetic | | ~30 GPa Static and Dynamic |
| Contact Numbers Controllability | Random Process | Deterministic Control | | |
| Surface Compressive Residual Stress (CRS) | ~1.0 GPa | | | ~2.0 GPa |
| Effective Depth of CRS | More than 1 mm | | | |
| Surface Hardness | Increase The effective depth is shallower than the effective depth of CRS | | | |
| Surface Roughness | Rougher | | | Smoother |
| Nano Structure Nano Twin | Nano Grain Refinement | | | |

*2.1. Water Jet Peening Process Which Was Applied to Section XI in Japan*

The WJP process utilizes cavitation bubbles to produce a shockwave which is generated in a submerged water jet as shown in Figure 1 [10].  The cavitation bubbles are produced by the strong shear force that acts on the boundary between the high-speed jet and the surrounding stationary water, and the bubbles are carried by the high-speed water jet to the material surface.  The collapse of the cavitation bubbles generates a large shock pressure more than 1 GPa that causes local plastic deformation.  Figure 1 shows a schematic of the process as applied to a flat plate and compressive residual stress mechanism.

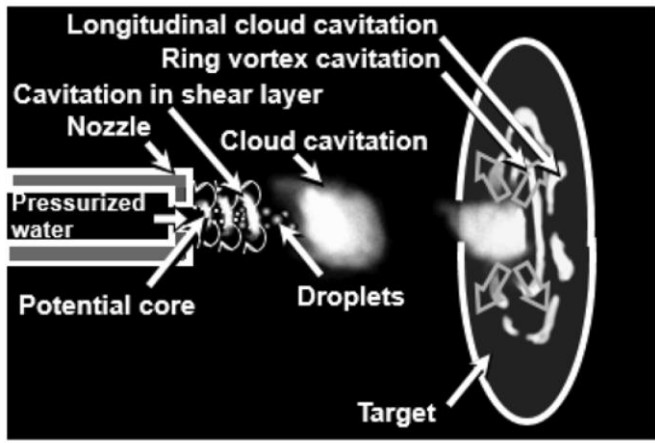

**Figure 1.** Principle of the WJP by submerged cavitating water jet.

By controlling the distance and the angle between the WJP nozzle and the metal surface, the collapse occurs on the metal surface and induces compressive residual stress, because plastic deformation generated by the intensive pressure wave is elastically constrained by the surrounding metal. The WJP process satisfied the requirement of ASME Code Case N-729-6 and N-729-5.

### 2.2. Under Water Laser Peening Process Which Was Applied to Section XI in Japan

The physical principle involved in laser peening treatment (also known as laser shock peening (LSP)) is considered to be an energy conversion procedure from a laser to shock wave that results in material plastic deformation by pressure of several GPa as shown Figure 2. After the passage of the shockwave, the permanent strain remains and the surrounding metal material constrains the strained region as a reaction to elastic strain, thus forming a compressive residual stress on the metal surface. Figure 3 shows the ULP setup for the underwater application simulating a mitigation operation with an inlet or outlet nozzle for the nuclear vessel. The ULP process satisfied the requirement of ASME Code Case N-729-6 and N-729-5.

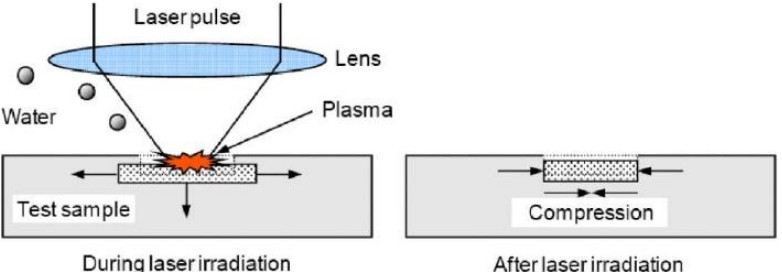

**Figure 2.** Fundamental mechanism of the LSP process.

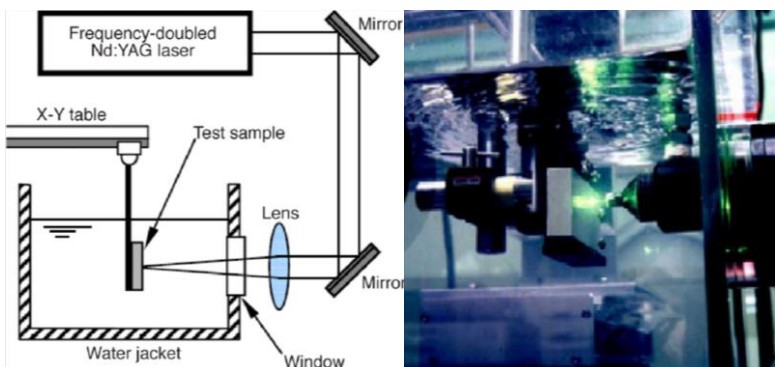

**Figure 3.** Experimental setup of the LSP process for underwater operation.

### 2.3. Air Laser Peening Process Which Is Being Prepared for Section XI and Section III

The fundamental mechanism of the ALP process is similar to the ULP. The ALP process uses a laser beam, typically in the range of 25 J per pulse and 25 ns duration with a resulting peak power of over $10^9$ watts, is imaged onto the surface of a material with a spot size in the range of 3 mm to 10 mm and, thus, gives irradiances of 11 GW/cm$^2$ and 1 GW/cm$^2$, respectively, as shown in Figure 4. Upon firing of a laser pulse, the intense electric field of the high-power laser ablates material at the ablation layer, creating a plasma that, within nanoseconds, reaches temperatures in the range of 16,000 K and lasts for about 2.5 times the laser pulse duration. The plasma would tend to "blow" off the surface of the substrate but is trapped between the substrate and the water layer, enabling the pressure to reach roughly 40,000 atmospheres. This rapid rise in surface pressure creates a shock wave with pressure above the yield strength of the substrate. The shock wave propagates through the ablative layer and into the metal, plastically deforming it as it propagates inward.

Some results of the PCRs of the ALP process for the requirement of the ASME Code Case N-729-6 and N-729-5 are presented in MRP-267-1. Technical data on the PCRs for ASME Sections III and XI Code case are being prepared in association with an international joint project.

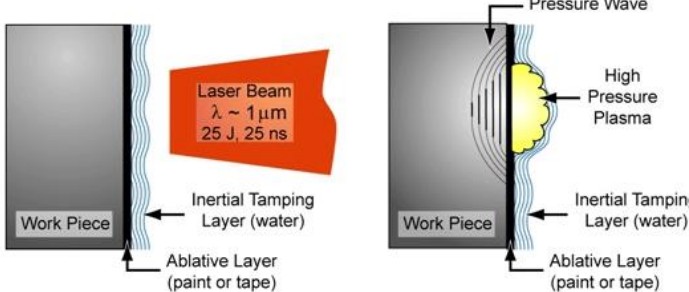

**Figure 4.** Schematic view of the LSP process showing the workpiece, ablative layer, inertial tamping layer, and laser input.

### 2.4. Ultrasonic Nanocrystal Surface Modification Which Is Being Prepared for Section XI and Section III

The mechanism of the UNSM process is to change the mechanical properties and microstructure by high-cycle, severe plastic deformation, elasto-plastic deformation, and elastic deformation. This deformation occurs at the surface and subsurface by up to 2.4 million strikes per minute and up to 10 million strikes per cm$^2$ with a pressure of up to 30 GPa. These strikes are induced by the resonance movement of an ultrasonic system as shown Figure 5. This device is very simple and small, hence it can easily be adapted to many kinds of machine tools and robots and even single-man manual operation is possible. The changes in mechanical properties and microstructure together with performance improvement of nuclear materials are summarized in Figures 6 and 7 [11].

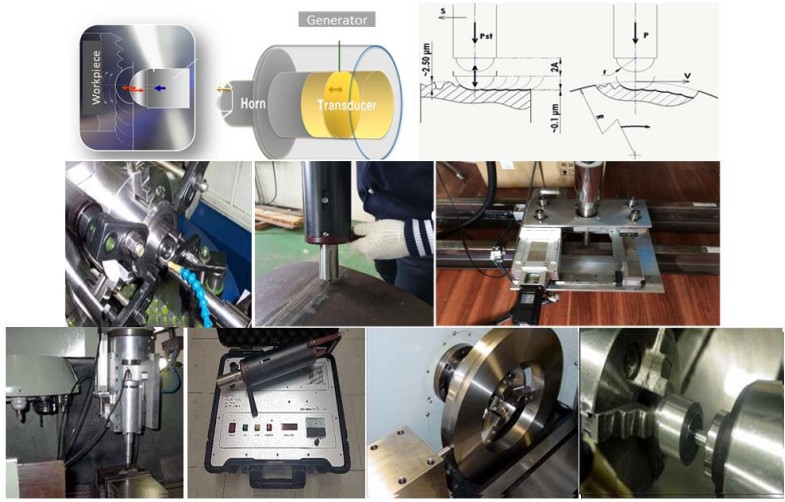

**Figure 5.** Mechanism and industrial applications of the UNSM process.

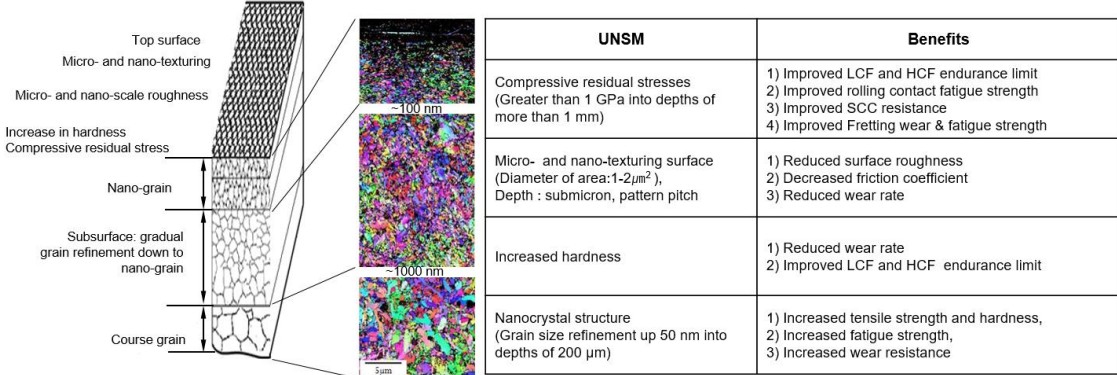

**Figure 6.** Changes in mechanical properties and microstructure and performance improvement.

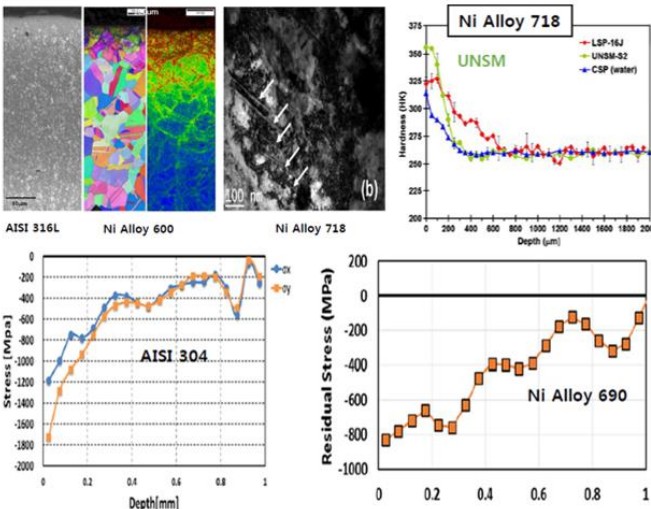

**Figure 7.** Examples of changes in mechanical properties and microstructure of nuclear materials.

The white papers for inclusion of the UNSM process into ASME Section XI Code Cases N-729-6 and N770-5 were presented at the ASME PVP 2017 Conference [11] and showed that the corrosion resistance was increased and the corrosion rate was reduced by the UNSM process, and the initiation of SCC was retarded by more than 1.5 times as shown Figure 8. Figure 8a,b show the SCC test results of stainless steel 316L-Ni Alloy 82 at 290 °C in 40% NaOH solution and at 340 °C in 0.01M $Na_2S_4O_6$ solution, respectively. Figure 8c,d show the anodic polarization test results of stainless steel 316L in 3.5% NaCl solution and the ASTM 262 Pr. C and double loop-electrochemical potentiokinetic reactivation (DL-EPR) test, respectively. Technical data on the PCRs for ASME Sections III and XI Code case are being prepared in association with an international joint project.

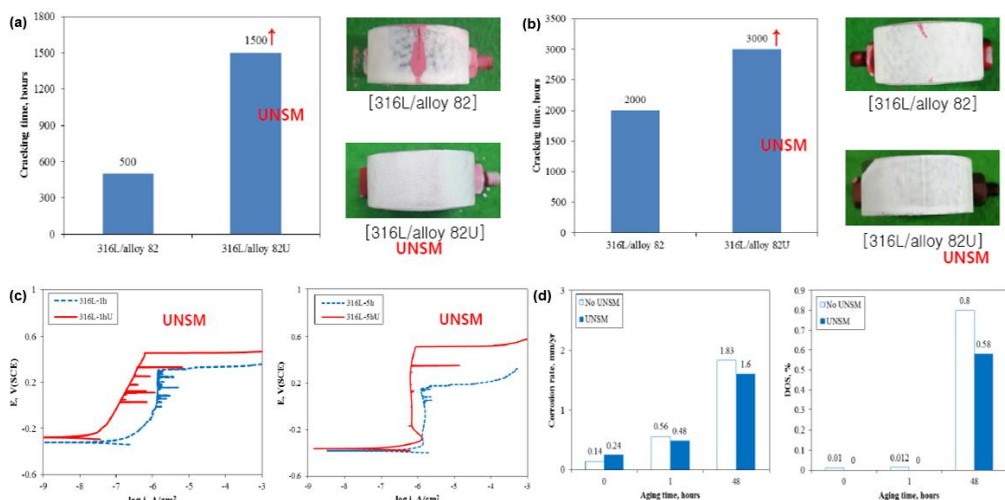

**Figure 8.** Improvements in SCC resistance (**a**), corrosion resistance (**b**), and corrosion rate of AISI 316L (**c**) and Ni Alloy 82 (**d**).

## 3. PCRs of the Advanced Surface Modification Process for ASME Code Case for Section XI and Section III and Technical Basis for Draft ASME Section III Code Case

### 3.1. PCRs of Advanced Surface Modification Process for ASME Code Case for Section XI and Draft for Section III

The titles of the peening performance criteria of the mandatory appendices of Code Cases N-770-5 and N-729-6 are "Performance Criteria and Measurement of Quantification Criteria for Mitigation by

Stress Improvement of Peening" and "Performance Criteria and Measurement or Quantification Criteria for Mitigation by Surface Stress Improvement (Peening) of the Reactor Vessel Upper Head Penetrations and Attachment Welds", respectively. The comparison between two code cases is summarized in Table 3.

**Table 3.** Comparison of performance criteria and mitigation process criteria: Section XI (N770-5 and N729-6) and New Code Case for Section III.

| Criteria | N770-5 Requirement | N729-6 Requirement | New Code case for Section III |
|---|---|---|---|
| ① Performance | The peening process will have resulted in a compressive stress in the susceptible material along the entire wetted surface under steady state operation (i.e., accounting for residual plus normal operating stresses) | | |
| | Peened surface will not exceed +10 ksi (+70 MPa) | | |
| ② Process Qualification | Analysis and demonstration testing shall be performed to quantify post-mitigation stress state and critical process parameters | | |
| | A 360 degree and 50% through wall weld repair must be assumed during pre-stress improvement state | Combined normal operating and residual stresses after peening shall not be greater than +10 ksi (+70 MPa) on the application surface | Ni alloy 690 has more resistance to SCC initiation than Ni alloy 600 |
| ③ Compressive Stress Field | The nominal depth of compression shall be demonstrated by testing | | |
| | At least 0.04 in (1.0 mm). Additional guidance for less than 0.04 in (1.0 mm) | Outside of nozzle and adjacent welds shall be no less than 0.04 in (1.0 mm), on the inside no less than 0.01 in (0.25 mm) | Effective depth is considered only when the undetected flaws exist. Its benefit is to slow down crack growing rate and it is not considered in Code Case N-770-5 and N-729-6 |
| ④ Mitigation Process | Effect produced by mitigation process shall remain effective for at least the expected remaining life of the component | | Effect produced by mitigation process shall remain effective for at least the target service life of the component |
| ⑤ Mitigation Process Qualification | Mitigation must be effective for service life as determined by analysis or demonstration with consideration of all applicable stresses (e.g., startup and shutdown, normal operating, thermal cyclic, transient stresses, residual stresses), load combinations, and relaxation | | |
| ⑥ Examination | Mitigation must not adversely affect ability to perform volumetric or surface examinations | Mitigation must not adversely affect ability to perform an ultrasonic testing (UT) examination | |
| ⑦ Ultrasonic Examination Qualification | Procedures, equipment, and personnel shall be qualified via blind demonstration on representative mockups | | |
| ⑧ Eddy Current Examination Qualification | Capability to perform eddy current examinations of the surface of relevant components will not be adversely affected | | |
| ⑨ Adverse Effects | Mitigation will not have an adverse effect on component or other components in the system | | |
| ⑩ Mitigation Process Geometry | Analysis or testing will be performed to verify no changes in component geometry | Mitigation will not have an adverse effect on component or other components in the system | |
| ⑪ Surface Effect | Verification that mitigation does not create undesirable/detrimental surface properties or conditions | | |
| ⑫ Inspectability | Surface or relevant volume will be inspectable via a qualified process | | |
| ⑬ Examination Coverage | Evaluation that examination coverages for all required surfaces and volumes can be obtained | | |
| ⑭ Existing Flaw Evaluation | If there are existing flaws, they will be addressed as part of the mitigation process via flaw examination and postulated flaw analysis | | Any flaw is not allowed in Section III |

The main goal is to minimize the likelihood of crack initiation, hence, the most important criterion in terms of performance and process is the surface residual stress after applying the advanced surface modification process. The maximum stress value after application of surface stress improvement is +70 MPa (N-729-6) or compressive (N-770-5) when considering operation pressure. The effective depth is to be considered for the undetected flaw by the examination process, so a minimum of 1 mm is for the welded area for both Code Cases and a minimum of 0.25 mm is for inner bore of tube for N-729-6 only. The effective depth in deterministic and probabilistic flaw analysis for N-770-5 is important for the consideration of inspection relief. However, surface cracks or flaws are not permitted in the examination process for Section III; therefore, the method for how to decide the effective depth could be re-evaluated. The other criteria of Code Cases N-770-5 and N-729-6 are the same or very similar except their different characteristics in the qualification process are due largely to the component configuration differences. The draft of PCRs for the new Code Case for Section III, which are under the consensus process, are also listed in Table 3.

### 3.2. Technical Basis on Advanced Surface Modification Process for ASME Section XI and III Code Case

The EPRI Technical Report MRP-267-R1 and EPRI Topical Report MRP-335-R3 provided the technical basis for ASME Code Case N-770-5 and N-729-6 for Section XI. The technical basis for a new Code Case developed for Section III applications are also being developed based on an EPRI Technical Report or developed white paper. Therefore, an international joint project between EPRI–Doosan–Dominion Engineering–Sun Moon University is underway to develop the data needed to support the technical basis document. Test specimens and mockups in this project were prepared as shown in Table 4. The measurement and analysis should satisfy the performance criteria and advanced surface modification process criteria of Table 3 and closely aligns to MRP-267-R1 and MRP-335-R3 as shown in Table 4 as well.

**Table 4.** Test and demonstration for technical basis: UNSM/ALP.

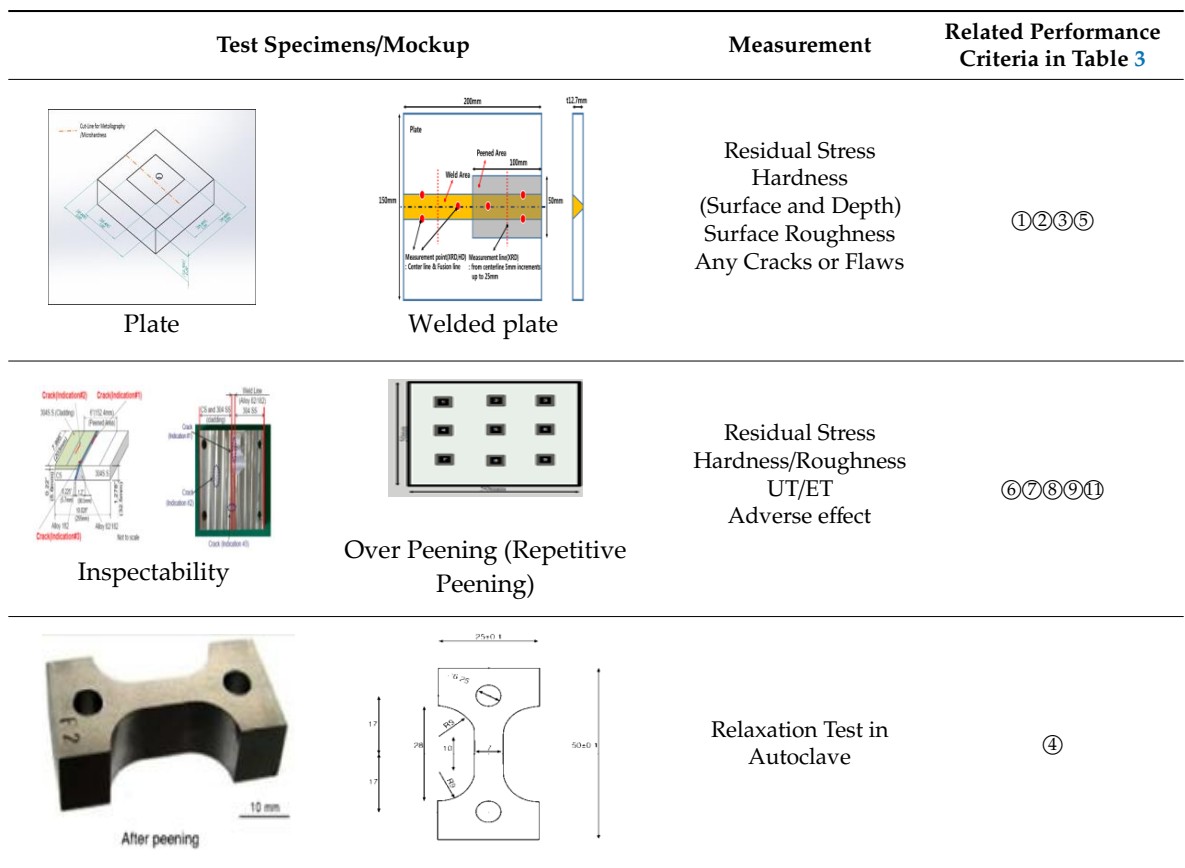

| Test Specimens/Mockup | | Measurement | Related Performance Criteria in Table 3 |
|---|---|---|---|
| Plate | Welded plate | Residual Stress Hardness (Surface and Depth) Surface Roughness Any Cracks or Flaws | ①②③⑤ |
| Inspectability | Over Peening (Repetitive Peening) | Residual Stress Hardness/Roughness UT/ET Adverse effect | ⑥⑦⑧⑨⑪ |
| After peening | | Relaxation Test in Autoclave | ④ |

**Table 4.** *Cont*.

| Test Specimens/Mockup | Measurement | Related Performance Criteria in Table 3 |
|:---:|:---:|:---:|
| 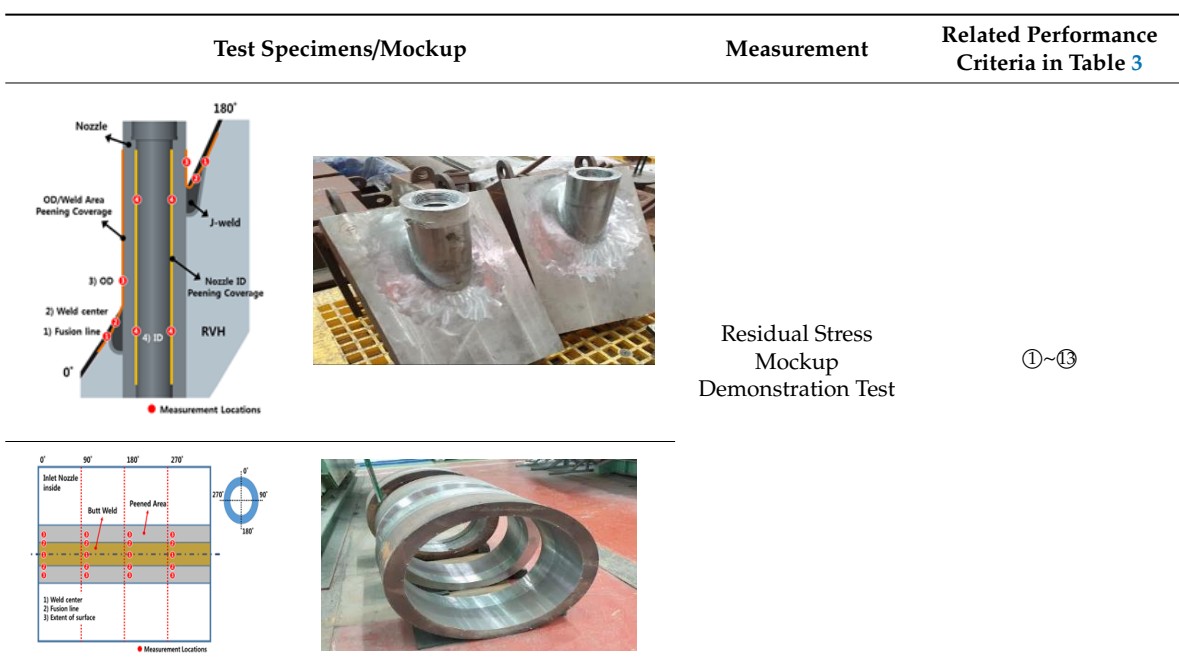 | Residual Stress Mockup Demonstration Test | ①~⑬ |

*3.3. Development of a Code Case for Application of Advanced Surface Modification Process*

3.3.1. Overview of Code Case Development Process

The ASME Code process utilizes volunteer members which includes input from interested stakeholders (utilities (US and International), United States Nuclear Regulatory Commission (USNRC), vendors, and consultants) to develop Code Rules to address industry needs. A Code Action is originated from a task group or working group in an ASME committee and requires review, comment, and subsequent consensus approval from multiple working groups and sub-groups. The number of other committees involved in the review, comment, and consensus approval process can vary significantly dependent upon the complexity and topic of the issue.

As the Code Action proceeds through the committee review process, all comments received from any reviewing committee level must be addressed and all changes to the existing Code (if necessary). The Action is presented again to the initiating committee for review and consensus approval before being presented again to all reviewing committees. This iterative process provides the checks and balances to ensure that all stakeholders have input into the proposed Action. These processes are summarized in Table 5. The new code case being developed for ASME Section III is currently at stage 2.

**Table 5.** Summary of Code Case development processes.

| | |
|:---:|:---|
| **Stage 1** | A stakeholder must identify a user need for change to the Code or Code Case supported by a technical basis to a task group or working group. |
| **Stage 2** | The proposed change must be accepted as a Code Action by a task group or working group and have an Action number assigned along with a program manager responsible for owning the Action throughout the process. |
| **Stage 3** | The review path needs to be determined, that is, what other working groups, sub-groups, and committees should review the proposed action as it ascends through the ASME approval process. |
| **Stage 4** | The ASME approval process starts at the task group or working group level to develop and comment on the technical basis supporting the change, along with the detailed word changes to the Code and to ensure that the proposed change does not conflict with the existing Code requirements. This includes compliance with all sections of the ASME Code and is not limited to ASME Sections III and XI. |

**Table 5.** *Cont*.

| Stage 5 | If questions of compliance to any section or paragraph of the ASME Code arise, the program manager is responsible for the resolution. |
|---|---|
| Stage 6 | The program manager must gain consensus acceptance of the task group and subsequent acceptance by each working group and sub-group having interest in the change. If consensus is not achieved or comments made requiring change to the Action, the Action is returned to the task group for resolution and re-approval. |
| Stage 7 | Following acceptance by the working groups and sub-groups, the program manager presents the proposed changes to the Standards committee for comment and approval. It should be noted that once the change is approved, the technical basis for the change is not published in the Code but resides with the Code Action. |

### 3.3.2. Development of a Code Case Draft for ASME Section III

An initial draft was developed, as shown Table 7, and is currently going through the iterative review and consensus process following Table 5. It will take a few years to be approved as a Code Case which will permit application of advance surface modification processes to new nuclear power plant components to prevent PWSCC or CISCC.

**Table 6.** Initial draft of the Code Case for the iterative review and consensus process.

Case N-XXX
Performance Criteria and Measurement of Quantification Criteria for Mitigation of Primary Water Stress Corrosion Cracking and Chloride Induced Stress Corrosion Cracking by Surface Stress Improvement for Section III, Division 1 and 3
Inquiry: What requirements may be used for Performance and Quantification Criteria of Surface Stress Improvement Process for Mitigation of PWSCC and CISCC in Section III, Division 1 and 3
Reply: It is the opinion of the Committee that the following requirements may be used
-1000　SCOPE
This Case was developed to apply Surface Stress Improvement Process to PWSCC and CISCC Susceptible Components Section III, Division 1 and 3
-1100　Components Subject to apply Surface Stress Improvement Process (SSIP)
(a)　PWSCC susceptible components: NCA-3250
(b)　PWSCC susceptible components: NB-4422, NC-4423.3, ND-4422, NG-4422
(c)　CISSCC susceptible components: NB-4422 (will be revised to division 3)
-1200　Surface Stress Improvement (SSI) Process (SSIP)
Any kinds of surface stress improvement processes, which can satisfy the performance criteria (2000) and sustainable CRITERIA of mitigation effects (3000) is to be used.
-2000　PERFORMANCE CRITERIA of Surface stress improvement process
To minimize the likelihood of crack initiation, the process will have resulted in a compressive stress in the repaired area of the susceptible material prior to consideration of operating stresses. The coverage of peened surface shall be extended at least 0.25 in (0.64 cm) beyond the weld repair area on the wetted surface. (Option 1)
To minimize the likelihood of crack initiation, the process will have resulted in a compressive stress in the susceptible material along the entire wetted surface or susceptible surface under steady-state operation. Susceptible material includes the weld, butter, and base material as applicable. The residual stress plus normal operating stress will be included in the evaluation. (Option 2)
-2100　Coverage and Depth of Effects (Option 2 continued)
An analysis, a demonstration test, or a combination of demonstration testing and analysis will be performed to confirm the surface stress improvement state.
(a)　The coverage of mitigated surface will be extended at least 0.25 in (0.64 cm) beyond the wetted surface in PWSCC.
(b)　The coverage of mitigated surface will be extended at least 0.25 in (0.64 cm) beyond the heat-affected zone (HAZ), tensional residual stress area, and crevice area at storage installation and during transportation in canister (CISCC).
(c)　The nominal compressive residual stress field will extend to a minimum depth of 0.04 in (1.0 mm) on the outside surface of the nozzle and attachment weld surface area susceptible to PWSCC initiation as defined in II-1000.
(d)　The nominal compressive residual stress field on the nozzle inside surface will extend to a minimum depth of 0.01 in (0.25 mm) on surfaces susceptible to PWSCC initiation as defined in II-1000.
(e)　The nominal compressive residual stress will extend to a minimum depth of 0.01 in. (0.25 mm) at (b).

**Table 7.** Initial draft of the Code Case for the iterative review and consensus process.

| |
|---|
| -2100　PROCESS QUALIFICATION CRITERIA |
| An analysis, a demonstration test, or a combination of demonstration testing and analysis will be performed to confirm the post-mitigation stress state. The testing will quantify the post-mitigation stress state exclusive of normal operating stresses. The testing will be used to demonstrate the critical process parameters and define acceptable ranges of the parameters needed to ensure that the required residual stress field (exclusive of operating stresses) has been produced on the mitigated surface. |
| -3000　MITIGATION PROCESS CRITERIA |
| The effect produced by the mitigation process will result in a peened surface with a stress state no greater than +10 ksi (+70 MPa) including residual and operating stresses. |
| -3100　MITIGATION PROCESS QUALIFICATION |
| An analysis, a demonstration test, or a combination of demonstration testing and analysis will be performed to confirm that the mitigation process maintains the compressive surface stress condition, normal operating and residual stress, for at least the remaining expected life of the component. The analysis or demonstration test plan will include startup and shutdown stresses, normal operating pressure stress, thermal cyclic stresses, transient stresses, and residual stresses. The analysis or demonstration test will account for: |
| (a)　load combinations that could relieve stress due to shakedown. |
| (b)　any material properties related to stress relaxation over time. |
| -4000　EXAMINATION CAPABILITY CRITERIA |
| The capability to perform examinations of the relevant volume or surface of the component will not have been adversely affected. |
| -4100　ULTRASONIC EXAMINATION QUALIFICATION CRITERIA |
| Ultrasonic examinations will be performed using personnel, procedures, and equipment qualified by blind demonstration on representative mockups that meet the requirements of -2500. Testing will be performed to demonstrate that the examination volume of the mitigated component can be examined subsequent to mitigation, including changes to component geometry, material properties, or other factors. |
| -5000　ADVERSE EFFECTS CRITERIA |
| Analysis or testing will be performed to verify the mitigation process does not cause erosion of surfaces, undesirable surface roughening, or detrimental effects in the transition regions adjacent to the peened regions. |
| -5100　MITIGATION PROCESS GEOMETRY EFFECTS CRITERIA |
| An analysis will have been performed to verify that the mitigation process does not result in changes to the component geometry that exceed Section III or original Construction Code design criteria. |
| -5200　SURFACE EFFECTS CRITERIA FOR MITIGATION BY PEENING |
| -Analysis or testing will have been performed to verify that peening does not cause undesirable hardness at the peened surface, erosion of surfaces, undesirable surface roughening, or detrimental effects in the transition regions adjacent to the peened regions. |
| -6000　INSPECTABILITY CRITERIA |
| The mitigated repaired region weld will be inspectable by a qualified process. |
| -7000　DEFINITIONS |

## 4. Concluding Remarks

Advanced surface modification processes, such as UNSM, ULP, ALP, and WJP, are promising solutions to improve the service life and safety of nuclear power plants and canisters. However, an ASME Code Case is a prerequisite condition to apply them in-service and considered a susceptible region mitigated to PWSCC or CISCC. Hence, all international stakeholders from the developers of the advanced surface modification process, utility companies, plant owners, engineers and constructors, maintenance workers to government regulators were assembled in a task group. The technical basis should be developed prior to the Code Case which could be used not only as technical basis but also as a practical guide to the application of the advanced surface modification process.

The PCRs of the advanced surface modification process for the technical basis of the ASME Code Case for Section XI were introduced and compared with the draft for the ASME Code Case for Section III. The design and plan of the test specimens and the mockup with measurement and performance criteria for developing the technical basis of the ASME Code Case for Sections III and XI were explained where UNSM and ALP were applied. The technical basis and draft Code Case of the advanced surface modification process for the ASME Code Case for Section III are being developed and will be published as either an EPRI Technical Report or white paper that could be used as a practical guide to the advanced surface modification process development or application. Considering the performance of

four advanced surface modification processes, it is expected that other processes could also satisfy the requirement of the ASME Code Case for XI and III.

**Author Contributions:** Conceptualization, Y.-S.P. and N.M.; methodology, Y.-S.P., S.C., A.A. and N.M.; validation, H.-U.H., M.A. and J.B.; formal analysis, Y.-S.P., S.C., A.A. and N.M.; investigation, Y.-S.P., S.C., A.A. and N.M.; data curation, Y.-S.P., S.C., A.A. and N.M.; writing—original draft preparation, Y.-S.P., S.C., A.A. and N.M.; writing—review and editing, Y.-S.P., S.C., A.A. and N.M.; supervision, Y.-S.P. and H.-U.H.; funding acquisition, Y.-S.P., N.M.; All authors have read and agreed to the published version of the manuscript.

**Funding:** This research was supported by external funding.

**Conflicts of Interest:** The authors declare no conflict of interest.

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
