# Peer review of "Application of the Advanced Surface Modification Process to the ASME Code Case for Sections III and XI of Nuclear Power Plants"

_metals, doi:10.3390/met10020210_

Round 1
Reviewer 1 Report
Page 3, Table 2
At the some cases of WJP, water column impacts are used. I recommend to correct “Cavitation bubble” to “Cavitation bubble / water column”.
Figure 3, Figure 1
Although Figure 1 was shown in EPRI Technical Report MRP-267, the explanation of cavitating jet is not correct. The figure was based on misunderstanding of cavitating jet structure, that shear layer around the cavitating jet directly hit the target. Note that the treatment area of the cavitating jet shows the ring area. Actual mechanism of a ring area induced by a cavitating jet is follows. Cavitation bubbles are generated in the shear layer around the jet; the cavitation bubbles combined together; they become a cloud; Cloud become a ring vortex; a part of ring vortexes are collapsed in a ring area.
Thus, I recommend to delete Figure 1 and switch to Figure 4 in the following reference.
Key Factors and Applications of Cavitation Peening, International Journal of Peening Science and Technology, Vol. 1, (2017), pp. 3 – 60.
Author Response
Thank you for your comments. Please find the attached file.

Reviewer 2 Report
An interesting manuscript challenging the advanced surface modification nanocrystal Surface Modification and Air Laser Peening to new nuclear power plants and new canister. The aim is to develop a technical basis and the Performance Criteria and Requirements for ASME Sections. However, the few comments are due:What demonstration test will be performed to confirm that the mitigation process has applied compressive stress in the materials. which x-ray imaging analysis is deployed to track the level of crack or mitigation. is there any image processing technique involved? the parameters measures and ranges to apply the mitigation's strength and preventive measures to surface erosion must be mentioned in 4100. code 6000 is very ambiguous and must be rewritten. this article must be cited in the references to fig. 7 and the SEM images: https://doi.org/10.1016/j.mtla.2019.100489 what method is used to decide the effective depth since the surface cracks or flaws are not permitted in the examination process for Section III? when the authors took my comments, I can reconsider my decision.
Author Response

(The authors gave the same response as above.)

Reviewer 3 Report
The present manuscript addresses an interesting and important overview on advanced surface modification processes.
Unfortunately, the content of the article is not in accordance with the requirements for a research article. The introduction does not provide sufficient scientific background, there are no adequate description of research design or research methods. The quality of figures can be significantly improved (are those just images from presentation slides?). The present content is not thoroughly discussed, resulting in the fact that the conclusions are very general and not directly connected to results.
Author Response

(The authors gave the same response as above.)

Round 2
Reviewer 2 Report
The manuscript is acceptable now.
Author Response
I have modified the style and spells. Native co-authors have had proof reading for this paper
Reviewer 3 Report
Dear authors, thank you for your reply.
If purpose of this paper is not to address an original research, it cannot be called "article" and presented in form of research paper.
In case you wanted to give an overview, please choose an another article type, which suitable for this aim (f.e. "review", "comment", "perspective", "technical note", etc.) and use a different article layout.
Author Response
Point 2: Open Review
(x) I would not like to sign my review report
( ) I would like to sign my review report
English language and style
( ) Extensive editing of English language and style required
( ) Moderate English changes required
( ) English language and style are fine/minor spell check required
(x) I don't feel qualified to judge about the English language and style
Comments and Suggestions for Authors
Dear authors, thank you for your reply.
If purpose of this paper is not to address an original research, it cannot be called "article" and presented in form of research paper.
In case you wanted to give an overview, please choose an another article type, which suitable for this aim (f.e. "review", "comment", "perspective", "technical note", etc.) and use a different article layout.
Response 2: Please provide your response for Point 2.(in red)
Yes, I have changed this paper from ‘article’ to ‘review’ as per your comment